# Unsupervised Learning for the Automatic Counting of Grains in Nanocrystals and Image Segmentation at the Atomic Resolution

**DOI:** 10.3390/nano14201614

**Published:** 2024-10-10

**Authors:** Woonbae Sohn, Taekyung Kim, Cheon Woo Moon, Dongbin Shin, Yeji Park, Haneul Jin, Hionsuck Baik

**Affiliations:** 1Seoul Center, Korean Basic Science Institute, Korea Road 22 6-7, Seoul 02841, Republic of Korea; swb1811@kbsi.re.kr (W.S.); rush98@kbsi.re.kr (T.K.); 2Department of Display Materials Engineering, Soonchunhyang University, 22 Soonchunhyang-ro, Sinchang-myeon, Asan 31538, Republic of Korea; cwmoon0810@gmail.com (C.W.M.); cbs08223@gmail.com (D.S.); 3Department of Chemistry, Research Institute for Natural Science, Korea University, Seoul 02841, Republic of Korea; yeeeji0610@gmail.com; 4Department of Energy and Materials Engineering, Dongguk University, Seoul 04620, Republic of Korea; hjin1@dongguk.edu

**Keywords:** unsupervised learning, Gabor filter, K-means clustering, automated image segmentation

## Abstract

Identifying the grain distribution and grain boundaries of nanoparticles is important for predicting their properties. Experimental methods for identifying the crystallographic distribution, such as precession electron diffraction, are limited by their probe size. In this study, we developed an unsupervised learning method by applying a Gabor filter to HAADF-STEM images at the atomic level for image segmentation and automatic counting of grains in polycrystalline nanoparticles. The methodology comprises a Gabor filter for feature extraction, non-negative matrix factorization for dimension reduction, and K-means clustering. We set the threshold distance and angle between the clusters required for the number of clusters to converge so as to automatically determine the optimal number of grains. This approach can shed new light on the nature of polycrystalline nanoparticles and their structure–property relationships.

## 1. Introduction

Mapping the grain distribution in polycrystalline nanoparticles is important to understanding the relationships among their structure, properties, and functionality. For instance, the grain size of gold nanoparticles significantly influences their optical properties [1]. Similarly, in the case of Fe_3_O_4_ nanoparticles, both the particle size and grain structure determine their magnetic properties [2]. Therefore, counting the number of grains and mapping their distribution is crucial for elucidating the physiochemical and functional properties of these nanoparticles. Transmission electron microscopy (TEM) and scanning TEM (STEM), with their atomic-scale resolutions, are powerful techniques for investigating the atomic structure and nanoscale grain distribution of nanoparticles [3,4,5,6]. Analyzing the grain distribution of nanoparticles by human eyes is laborious and is associated with bias in the determination process. Therefore, manual grain-number counting and grain segmentation are far from being accurate [7,8].

To overcome the limitations of manual TEM image analysis, automated methods utilizing machine learning (ML) and deep learning (DL) have emerged as techniques for an automated analysis; these methods can process and analyze large volumes of TEM and STEM images with high accuracy and efficiency [9]. ML, including supervised and unsupervised learning, has been employed in image processing and particle segmentation [10,11,12,13,14]. In the case of particle segmentation, ML helps define the shape, size, and crystallographic orientation of nanoparticles [15,16]. Recently, Xu et al. developed a U-net-based DL model for an automated analysis of the grain morphology from TEM images [17]. The DL model automatically extracted relevant features from raw data, making it highly effective for identifying and quantifying grain structures. Applying ML and DL for (S)TEM image analysis can not only reduce the labor and time required for manual examination but can also significantly enhance the precision and consistency of the results. Furthermore, these automated techniques can continuously improve the performance by learning from new data, thus ensuring adaptability to various types of nanoparticles and imaging conditions.

Despite their advantages, ML and DL methods demand a large number of labeled datasets for training to achieve high accuracy and reliability. Obtaining such extensive datasets can be challenging and time consuming because it requires manual annotation by experts. Moreover, the performance of these models is limited by the quality and diversity of the training data. Insufficient or biased datasets can lead to inaccurate predictions and limit the generalizability of models to new, unseen data. Although a DL-based microscopy image analysis provides good performance, it cannot sufficiently explain the process of producing the analysis results. In the meantime, Bárcena-González et al. succeeded in the segmentation of polycrystalline nanoparticles using unsupervised learning, but the work showed limit potentials and the methodology still needs automation [18].

In this study, we applied a Gabor-filter-based unsupervised clustering method to count the number of grains and map the grain distribution in polycrystalline nanoparticles. Unsupervised clustering involves constructing a feature matrix using filtered images and non-negative matrix factorization (NMF) of the feature matrix, followed by the K-means clustering. We set the threshold value of the distance between clusters for automated segmentation. In comparison with previous ML-based segmentation methods, we used only 252 filtered images as a dataset. Moreover, this methodology not only divides grains with different crystallographic orientations but also shows surface-reconstructed regions (particularly the Pt-skin layer of PtNi intermetallic nanoparticles) as independent features. 

## 2. Materials and Methods

### 2.1. Gabor Filter

The Gabor filter, developed by the Nobel Prize-winning physicist Dennis Gabor in 1971, is a linear filter that is widely used in image processing for texture analysis. It is particularly effective in edge detection, fingerprint extraction, and pattern recognition. Since Gabor’s pioneering work, this technique has evolved significantly, enabling phase retrieval to achieve sub-nanometric resolution using high-resolution transmission electron microscopy (HRTEM). The phase information extracted from electron micrographs provides insights into the physical properties of a sample, such as its electrostatic and magnetostatic potentials and strain, which are typically analyzed using Fourier space methods [19,20,21,22,23].

Gabor discovered that the uncertainty principle is relevant to signals, revealing that it is impossible to accurately determine a signal’s exact location in both the frequency and time domains at the same time. This introduces a trade-off between the time and frequency resolutions, with a minimum bound on the combined product. Gabor demonstrated that the most general function achieving this minimum is a Gaussian-shaped envelope multiplied by a complex sinusoid, which maximizes the accuracy in the time–frequency domain. Originally, the Gabor filter was designed as a linear filter for 1D signal analyses. Daugman later expanded this concept to two dimensions, significantly enhancing its usefulness for texture analysis, feature extraction, edge detection, image compression, and various other image-processing applications. In two dimensions, a Gabor filter is typically represented as a sinusoidal Gaussian function. In the spatial domain, a 2D Gabor filter is defined as a Gaussian function modulated by a complex sinusoidal plane wave. These are expressed as follows:
g (x, y) = w (x, y) ∗ s (x, y),(1)
(2)$$w\;(x,\;y)=\frac{1}{\sqrt{2\pi \sigma }}e^{-\frac{1}{2}(\frac{x^{2}}{\sigma_{x}^{2}}+\frac{y^{2}}{\sigma_{y}^{2}})}$$
where *σ* represents the standard deviation of the Gaussian function, and $\sigma_{x}^{2}$ and $\sigma_{y}^{2}$ are the variances along the *x*-axis and *y*-axis, respectively, defining the width of the major and minor axes of the Gaussian envelope. The parameters *u*_0_ and *v*_0_ define the spatial frequency of the sinusoid, and *φ* determines its phase. Since a Gabor filter is the product of a Gaussian function and a sinusoid, its Fourier transform is the convolution of their individual transforms. This results in a Gaussian centered at the frequency of the harmonic function associated with the sinusoid. This characteristic is crucial for understanding why Gabor filters are highly effective in feature extraction based on fringe orientation and spacing, making them an invaluable tool for analyzing the electron microscopy images of crystalline materials.

The Gabor filter can be expressed in a simplified form as a combination of two components: a Gaussian function and a complex exponential. The Gaussian part controls the spatial extent, while the complex exponential introduces oscillations. The parameters in the filter include *σ*_*x*_ and *σ*_*y*_ which define the spread in the x and y directions, respectively; *θ*, which represents the orientation of the filter; *γ*, the aspect ratio; *λ*, the wavelength of the sinusoidal component; and *ϕ*, the phase offset. Additionally, the variables *x*′ and *y*′ represent rotated coordinates, with *x*′ being a combination of *x* and *y* based on the angle *θ*, and *y*′ being a similar combination but with a different sign for the sine component. These rotated coordinates adjust the filter’s response based on its orientation in space. 

The critical parameters of a Gabor filter are the orientation angle (*θ*) and the wavelength (*λ*), which determine the spacing between the fringes and, thus, the filtering channel. The impact of varying the wavelength is shown in Appendix A, where *λ* directly influences the width of the filter channel. Appendix A show changes in the orientation of a Gabor filter. To apply Gabor filtering to an image *f*(*x*,*y*), it is convolved using the Gabor filter.

Gabor filtering accentuates the local texture that matches the orientation and wavelength of the filter. By adjusting the orientation, the filter can pass features that are aligned in a specific direction. Similarly, altering the wavelength highlights the textures with fringe patterns of different spacings. The parameters *σ*_*x*_ and *σ*_*y*_ control the bandwidth of the 2D filter, defining the size of the image region contributing to a pixel value in the filtered image.

A Gabor filter bank can be designed to extract significant features from an image, comprising a set of Gabor filters with various orientations and spacings, typically covering the entire spatial and orientation spectra. As shown in Appendix A, the output from this filter bank characterizes each pixel in the original image based on fringe patterns in its vicinity.

### 2.2. Non-Negative Matrix Factorization 

NMF is a dimensionality reduction technique used for data analysis. It decomposes a non-negative matrix *V* into a product of two lower-dimensional non-negative matrices *W* and *H*. This method is particularly effective for extracting meaningful features from high-dimensional datasets, such as in image processing and text mining.

### 2.3. K-Means Clustering

A common task in data analysis is to identify groups of similar data points, often referred to as clusters, in an unsupervised manner without prior labeling of the data. Clustering algorithms group sets of items based on their similarities. In this study, the K-means clustering method was used, with the Euclidean distance serving as the similarity metric. K-means is a straightforward and widely used clustering approach. The algorithm is described as follows:

Let K represent the desired number of clusters, and a dataset of *N* points, each described by M features, is considered.
(*x*_*i*_ ∈ *R*^*M*^, *i* = 1…*N*)

K initial centroids (often chosen from the data points) are randomly defined.

This is repeated until a stopping criterion is met: (a) assigning each item to the nearest centroid based on the Euclidean distance, and (b) recalculating the centroids as the mean of all points assigned to each centroid.

In the method presented herein, *N* corresponds to the number of pixels in the analyzed image, and *M* corresponds to the number of Gabor features per pixel, which are determined by the number of orientations and spacings used in the filtering process. Ultimately, each pixel is classified into one of the K clusters based on its similarity to the others, as determined by its Gabor features. A primary limitation of the K-means algorithm is that it cannot guarantee convergence to a global optimum, because the final result depends on the initial centroids chosen at random. However, as the algorithm is typically fast, it is commonly run multiple times with different initializations. In our experiments, convergence was consistently achieved without any problems.

### 2.4. Synthesis of Au Nanoparticles

Gold (III) chloride hydrate salt (HAuCl_4_·3H_2_O) (99%) was purchased from Sigma-Aldrich. Anhydrous sodium citrate (99.5%) was purchased from Daejung Chemicals. A solution of 20 mL HAuCl_4_ (1.0 mM) was placed on a hot plate in a glass jar (100 mL) and heated until the solution temperature was reached at 100 °C. Thereafter, 2 mL of sodium citrate solution (40 mM) was rapidly injected into a glass jar. During the synthesis protocol, the vial was sealed to prevent water evaporation. Anhydrous sodium citrate was used as a reducing agent. After the injection, the solution turned dark blue. Finally, after a few minutes, a red wine color appeared, which is evidence of the synthesized gold nanoparticles [24]. The solution was maintained at 100 °C while stirring for 30 min to finish the nanoparticle growth process [25].

### 2.5. Synthesis of PtNi Intermetallic Nanoparticles

To produce the optimal PtNi intermetallic nanoparticles, a two-step annealing process was utilized. Initially, the precursor was heated to 1100 °C at a rate of 5 °C per minute for 2 h. This was followed by a second heating stage at 550 °C for 12 h in a 5% H_2_/Ar atmosphere, after which the material was allowed to cool naturally to room temperature [26].

### 2.6. Synthesis of PtCo Intermetallic Nanoparticles

H_2_PtCo_6_∙H_2_O (chloroplatinic acid hexahydrate) and CoCl_2_∙6H_2_O (cobalt(II) chloride hexahydrate) were purchased from Sigma-Aldrich from Seoul, Republic of Korea. All the chemicals were used as received without further purification.

PtCo intermetallic nanoparticles were synthesized using an impregnation reduction method followed by thermal annealing. In a typical synthesis of the PtCo intermetallic nanoparticles, H_2_PtCo_6_∙H_2_O 0.0669 g (0.135 mmol), CoCl_2_∙6H_2_O 0.0214 g (0.09 mmol), carbon support (Vulcan XC-72, Naracelltech from Seoul, Republic of Korea) 0.08 g, and 100 mL of DI water were prepared in a 250 mL round-bottom flask (RBF). To achieve a homogeneous dispersion, the mixed solution underwent sonication in an ice bath for 1 h. Subsequently, the RBF was placed directly in a preheated oil bath at 90 °C for 4 h. During this process, all the deionized water evaporated, leaving behind a powder mixture of the carbon support and metal precursor. Finally, the resulting composite was thermally annealed at 600 °C for 2 h using a tube furnace, with the annealing temperature increasing by 1 °C per 1 min from room temperature to 600 °C.

### 2.7. STEM Characterization and Simulation of Nanoparticles

To obtain high-angle annular dark-field (HAADF) STEM images, an FEI double Cs-corrected Titan Themis transmission electron microscope was operated at 300 kV. A STEM simulation was performed using a Dr. Probe (Version 1.11.0) [27].

## 3. Results and Discussion

### 3.1. Segmentation of Polycrystalline Nanoparticles

Figure 1 illustrates the methodology. First, multiple Gabor filters are defined. Thirty-six different orientations and seven wavelengths were set, and 252 filtered images were generated. Each filtered image shows how much of the nanoparticle contains faces in a particular orientation. A feature vector was constructed using filtered images such that the matrix has a size of 252 rows and 512 × 512 = 262,144 columns (images have a pixel size of 512 × 512). NMF was applied to the feature matrix, followed by K-means clustering of the dimensionally reduced feature vector. K-means clustering allows for clustering of grains that contain planes with the same orientation and interplanar distance within a nanoparticle. A clustered matrix comprising class numbers and different colors was assigned to different class numbers. When various types of nanoparticles are applied to the methodology, the segmentation results are sensitive to the parameters of Gabor-filter. Hence, different values of optimal parameters had been set.

The Gabor-filter-based unsupervised methodology was first applied to a simulated HAADF image of an Au nanoparticle with a five-fold twin.

We constructed an atomic model (see Appendix A) and simulated the HAADF-STEM image using Dr. Probe. Figure 2a illustrates a simulated HAADF-STEM image of the Au nanoparticle with five-fold twins; Figure 2b–e illustrate the image segmentation results for four different *k* values. Ground truth segmentation result is included as shown in Figure 2f. We found that *k* = 6 yielded the best segmentation result, which shows the most similarity to the ground truth. Appendix A shows each segmentation result with various k values, compared with the ground truth by evaluating image similarity. As expected, segmentation result with k = 6 shows the best similarity result. Moreover, surface and boundaries between phases and grains are assigned as grey lines. The segmentation results demonstrated that applying the NMF can help effectively divide grains with different orientations. The reliability of the methodology is verified by a comparative test with other clustering methods, as shown in Appendix A. NMF followed by the K-median method (see Appendix A) and the K-means method without NMF (see Appendix A) are used for comparison, showing that the suggested clustering method provides the best segmentation results.

Next, the experimental image was segmented. Figure 3a illustrates a high-resolution HAADF-STEM image of a twinned Au nanoparticle coalesced with a smaller one; Figure 3b–e illustrate the image segmentation results with four different *k* values. We found that NMF is an appropriate method for image segmentation. Such a dimension reduction method can help cluster the feature matrix and perform semantic segmentation of polycrystalline nanoparticles. Note that the grain number “7” is classified as the background when *k* = 7 (see Figure 3b). Since there is no clear lattice fringe in the grain “7”, it is possible to cluster it as the background.

The algorithm converges for *k* = 10. The counting of grains was successful; however, the segmentation of the grains along the boundaries was not precise. This issue was particularly evident in grains 4 and 5, as shown in Figure 3a, where the grains exhibited irregular shapes, resulting in imperfect boundary delineation.

### 3.2. Potential of the Methodology: Capturing Unknown Features

To extend the scope of the unsupervised clustering method to images of nanoparticles with more complicated microstructures, we applied image segmentation to the HAADF-STEM image of PtNi intermetallic nanoparticles with more than two different crystallographic grains. For PtNi, approximately 2–3 atomic layers of Pt were wrapped around the intermetallic phase. The segmentation results with *k* = 6 show that not only the grains but also the surface and interface regions, marked as orange and red regions, respectively, are independent classes (See Figure 4b). Because Gabor filters with seven different wavelengths were used for the filtering process and the lattice parameters of the Pt skin layer and PtNi intermetallics were different, they were classified into different classes of clusters. The segmentation process revealed the heterogeneity within the PtNi intermetallic nanoparticles, highlighting regions with distinct textural characteristics, which is crucial for understanding the microstructure and catalytic property relationship of the material. The electrochemical properties of PtNi intermetallic nanoparticles are affected by the distribution of the Pt-skin layers. By employing advanced image-processing techniques, we can quantitatively assess the structural differences and gain insights into the behavior of the material under various conditions. This detailed segmentation and classification provides a more comprehensive understanding of nanoparticle composition and can be useful in future material design and optimization efforts.

Our methodology can distinguish and segment grains that cannot be visualized manually. Figure 5a shows a HAADF-STEM image of the PtCo intermetallic nanoparticle, where the grain distribution is indiscernible to the naked eye. In comparison, as shown in Figure 5b, our methodology can automatically distinguish between the phase and grain distributions. The yellow region corresponds to the surface region, whose atomic arrangement and lattice parameters are not equal to those of the PtCo intermetallic phase; thus, it is classified as a different cluster. Thus far, the phase segmentation of intermetallic nanoparticles has been achieved through 3D reconstruction using electron tomography [28,29,30,31,32]. This technique requires a considerable number of rotational images, and because nanoparticles tend to undergo phase transformation or degradation under electron-beam irradiation, this process is time-consuming. In contrast, our methodology can schematically show the grain and phase distributions without providing detailed information about the phases of the nanocrystals.

### 3.3. Automated Segmentation of Nanoparticles

Unsupervised clustering with different *k* values is limited because it cannot automatically determine the number of grains in a nanoparticle. This shortcoming arises from the dependency of the k-means algorithm on the person determining the optimal *k* value. In addition, our methodology assigned a class of other nanoparticles from the image that were not of interest, as illustrated in Appendix A. When the segmentation of nanoparticles is performed, conventional K-means clustering requires a manual determination of the optimized *K* value, and it cannot distinguish between the nanoparticles, which is the research of interest, and other nanoparticles with low crystallinity. Here, we set threshold values for the distance between clusters such that the number of clusters converges, even with high *k* values. Figure 3 illustrates the results of clustering with the threshold values, which enable automated segmentation. Compared with Appendix A, the automated segmentation helped classify low-crystallization nanoparticles of interest into a class similar to the background (see Figure 6g,h). The proposed methodology addresses the limitations of conventional k-means clustering by introducing threshold values for distances between clusters, thereby enabling automated segmentation. This approach not only converges the number of clusters at high *k* values but also effectively distinguishes between nanoparticles of interest and other nanoparticles with low crystallinity, thus improving the accuracy and efficiency of nanoparticle segmentation. To show the computational complexity, a pseudo-code schematization is included in the Appendix A.

## 4. Conclusions

We developed a methodology for the automated segmentation of polycrystalline nanoparticles based on (1) the application of a series of Gabor filters for the powerful feature extraction of grains, (2) unsupervised clustering for particle segmentation, and (3) automation of clustering by setting the threshold value. We not only demonstrated the automated segmentation of the nanoparticles but also revealed hidden features using this methodology even with relatively small number of training datasets. The unsupervised clustering method for the segmentation can help visualize the phase distribution of complex intermetallic nanoparticles, thus enabling the quantitative analysis of multigrain/phase nanoparticles, such as the calculation of the ratio of individual grains.

We believe that this methodology can not only provide crystallographic information but also new shed light on the structure–property relationship of nanocrystals.

## Figures and Tables

**Figure 1 nanomaterials-14-01614-f001:**
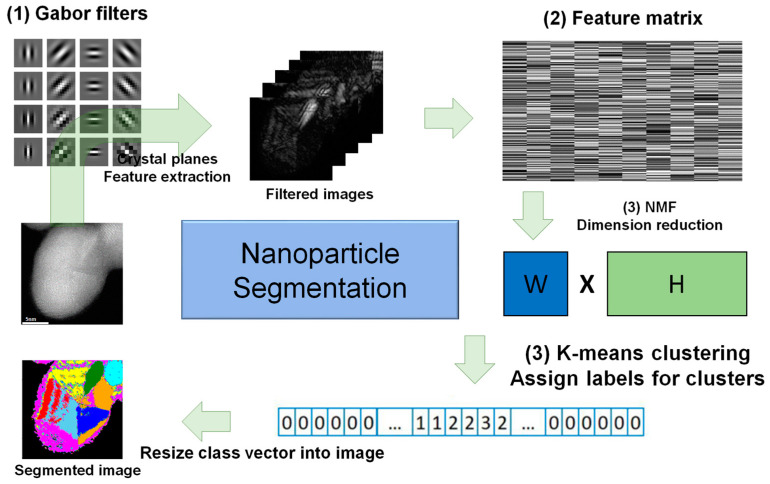
Schematic of the Gabor-filter-based clustering for particle segmentation. (1) Application of multiple Gabor filters, (2) creation of a feature vector for each pixel to obtain a feature matrix, and (3) dimension reduction using NMF followed by K-means clustering. The class vectors are rearranged into a 2D matrix, illustrating the segmented image.

**Figure 2 nanomaterials-14-01614-f002:**
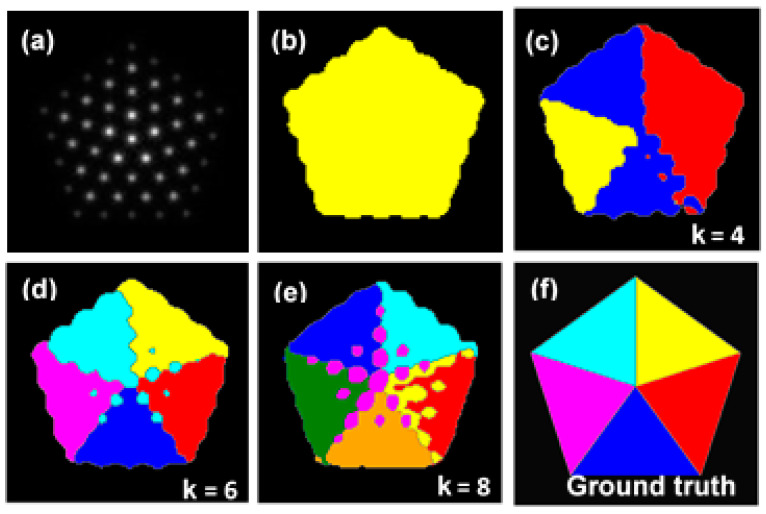
Sequence of *k* in the Au nanoparticles with five-fold twin and colorized segmentation, which are compared with the ground truth. (**a**) HAADF-STEM image, showing five-fold twins of the particle. Segmentation and colored classes for (**b**) *k* = 2; (**c**) *k* = 4; (**d**) *k* = 6; (**e**) *k* = 8. (**f**) Ground truth of the segmentation of (**a**). The different colors indicate the different classes after clustering.

**Figure 3 nanomaterials-14-01614-f003:**
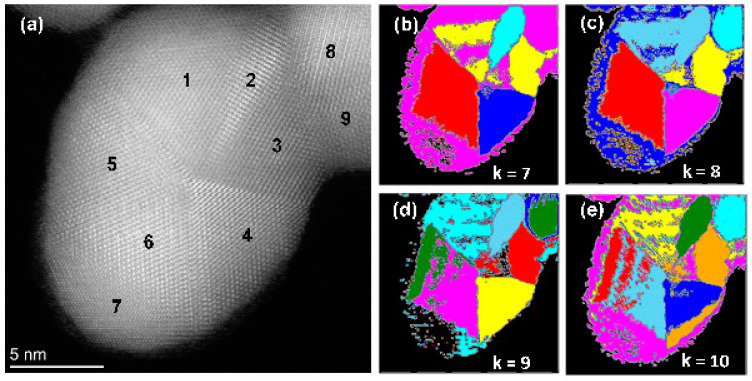
Segmented images of the Au nanoparticles with various *k* values. (**a**) HAADF-STEM image, showing five-fold twins of the particle. Segmentation and color maps for (**b**) *k* = 7; (**c**) *k* = 8; (**d**) *k* = 9; (**e**) *k* = 10. The different colors indicate the different classes after clustering.

**Figure 4 nanomaterials-14-01614-f004:**
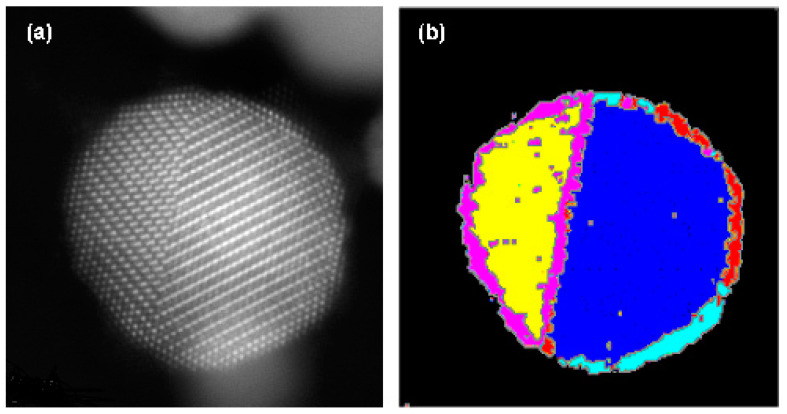
Segmentation of PtNi intermetallic nanoparticles. (**a**) HAADF-STEM image of PtNi intermetallic nanoparticle; (**b**) segmented image with *k* = 5. The different colors indicate the different classes after clustering.

**Figure 5 nanomaterials-14-01614-f005:**
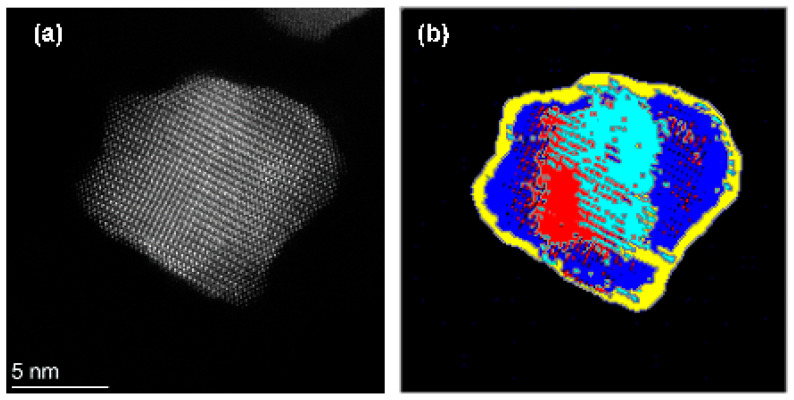
Segmentation of the PtNi intermetallic nanoparticles. (**a**) HAADF STEM image of the PtNi intermetallic nanoparticle; (**b**) segmentated image with k = 6. The different colors indicate the different classes after clustering.

**Figure 6 nanomaterials-14-01614-f006:**
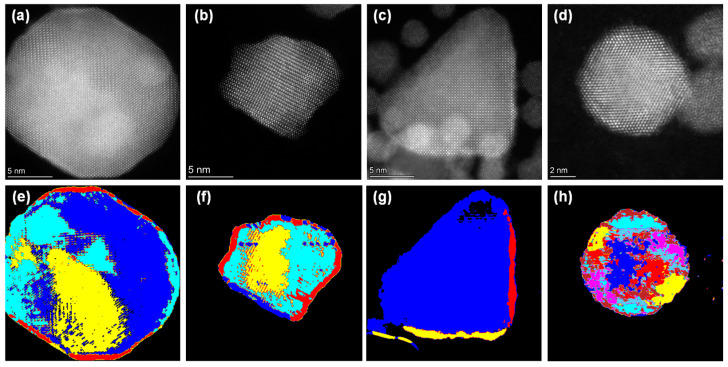
Automated segmentation by setting threshold value with *k* = 10. (**a**–**d**) HAADF-STEM images of intermetallic nanoparticles for image segmentation. With the same *k* values, those images are segmented with optimal *k* values of (**e**) 5, (**f**) 5, (**g**) 4, and (**h**) 9. The different colors indicate the different classes after clustering.

## Data Availability

The data presented in this study are available upon request from the corresponding author.

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
