# Peer review of "Unsupervised Learning for the Automatic Counting of Grains in Nanocrystals and Image Segmentation at the Atomic Resolution"

_nanomaterials, 2024, doi:10.3390/nano14201614_

Round 1

Reviewer 1 Report

Comments and Suggestions for Authors

The goals of this research are significant and are clearly and comprehensively stated in the introductory section. However, a description of related works and a thorough discussion of the limitations and critical points of these works are missing. I recommend that this discussion be placed in a paragraph preceding section 2.

Furthermore, a figure showing the architectural scheme of the method and a discussion of its functional components is missing.

The schema in Fig.1 is not significant. It would be appropriate to include a flow diagram showing how the methods used (Gabor filter, K-means etc.,) are connected to each other.

It is appropriate to include a pseudocode schematization of the algorithm and discuss its computational complexity.

Comparative tests with computational intelligence related methods must be added, otherwise it is not possible to evaluate the performance efficiency of the proposed method.

Reviewer 2 Report

Comments and Suggestions for Authors

In this study, the authors focus on the use of unsupervised learning methods to count and segment grains in polycrystalline nanoparticles. The methodology involves applying Gabor filters to high-resolution electron microscopy images, followed by dimension reduction using NMF and clustering with K-means. This approach helps to automatically segment and count grains, which is vital for understanding structure-property relationships in nanocrystals. 

The study makes use of unsupervised learning, specifically Gabor filters, NMF and K-means clustering, to automate the counting and segmentation process of nanocrystals. This eliminates manual bias and improves efficiency. The approach has been successfully tested on both real and simulated HAADF-STEM images with robust results. This methodology can be extended to a variety of nanomaterials, including complex intermetallic nanoparticles, indicating a wider range of applications in materials science.

However; 

1- The dependence of K-means clustering on the initial value of K, which must be manually adjusted to achieve the optimal segmentation. This contradicts the claim of full automation and should be clarified.

2- The study is based on a relatively small dataset of 252 images for training, which may limit the generalizability of the model to more diverse and larger datasets.

3- The segmentation results struggle to accurately detect boundaries between grains, especially those with irregular shapes, which should be explained.

4- The performance of the algorithm is sensitive to various input parameters, including filter settings and threshold values, which may require fine-tuning for different nanoparticle types.

However, it introduces an innovative and practical approach to automated nanoparticle grain segmentation and counting that has the potential to make significant contributions to the field of materials science. I kindly request the authors to clarify the points I have kindly raised above. 

Reviewer 3 Report

Comments and Suggestions for Authors

In this paper, the author developed an image processing pipeline for high-angle annular dark-field scanning transmission electron microscopy (HAADF-STEM) images to automatically segment grains in polycrystalline nanocrystals. The author synthesized the nanocrystals and captured atomic-resolution HAADF-STEM images of the samples. Gabor filtering and feature extraction were applied to the captured images, and the resulting feature matrix was then used for clustering image pixels into grain indexes.

[General comments]

The concept of the paper is simple and straightforward. The author discussed the limitations of k-means clustering well, particularly noting that "it cannot automatically determine the number of grains in a nanoparticle." To further improve this work, it would be beneficial to explore methods for automatically determining the value of k for clustering, as this would make the approach more suitable for a fully automated workflow. However, I understand this is a challenging task. Thus, I recommend the paper for publication with minor corrections, without requiring additional experiments.

[Minor issues]

All of the figures appear to demonstrate segmentation using the author's chosen k value. It would be helpful if the author could include their own human-segmented results, as a ground-truth comparison.

Round 2

Reviewer 1 Report

Comments and Suggestions for Authors

The authors have taken into account all my suggestions. A figure has been inserted that schematizes the architecture of the method and a pseudocode description of the algorithm and the results of comparative tests have been added. I consider this paper publishable in this version.